# A Semantic Similarity-Based Identification Method for Implicit Citation Functions and Sentiments Information

Rami Malkawi [1,*] , Mohammad Daradkeh [1,2] , Ammar El-Hassan [3] and Pavel Petrov [4]

1   Faculty of Information Technology and Computer Science, Yarmouk University, Irbid 21163, Jordan
2   College of Engineering and Information Technology, University of Dubai, Dubai 14143, United Arab Emirates
3   Computer Science Department, Princess Sumaya University for Technology, Amman 11195, Jordan
4   University of Economics-Varna, 9000 Varna, Bulgaria
*   Correspondence: rmalkawi@yu.edu.jo

**Abstract:** Automated citation analysis is becoming increasingly important in assessing the scientific quality of publications and identifying patterns of collaboration among researchers. However, little attention has been paid to analyzing the scientific content of the citation context. This study presents an unsupervised citation detection method that uses semantic similarities between citations and candidate sentences to identify implicit citations, determine their functions, and analyze their sentiments. We propose different document vector models based on TF-IDF weights and word vectors and compare them empirically to calculate their semantic similarity. To validate this model for identifying implicit citations, we used deep neural networks and LDA topic modeling on two citation datasets. The experimental results show that the F1 values for the implicit citation classification are 88.60% and 86.60% when the articles are presented in abstract and full-text form, respectively. Based on the citation function, the results show that implicit citations provide background information and a technical basis, while explicit citations emphasize research motivation and comparative results. Based on the citation sentiment, the results showed that implicit citations tended to describe the content objectively and were generally neutral, while explicit citations tended to describe the content positively. This study highlights the importance of identifying implicit citations for research evaluation and illustrates the difficulties researchers face when analyzing the citation context.

**Keywords:** citation text identification and classification; implicit citations; citation content analytics; sematic similarity; term frequency-inverse document frequency (TF-IDF); vector space model (VSM)

## 1. Introduction

Research publications are by no means self-contained, isolated entities but rather individual pieces of literature that relate to previous research. This linkage between research publications is established through the use of citations, which serve as a bridge between the citing document and the cited document. This linkage of citation data has long been used to analyze scientific knowledge, identify disciplines, and predict future scientific directions. However, as citation analysis becomes more sophisticated, its impact on knowledge generation and dissemination is increasingly scrutinized [1]. In addition to earlier approaches that have evaluated research based on peer review, existing methods such as the h-index and journal impact factors (JIFs), which measure the impact of citations, have been used to evaluate research. The use of citation counts as a typical measure of the scientific impact of a research publication, researcher, or institution has been severely discredited in recent studies [2]. In citation analysis methods that determine the scientific impact of publications based on the number of citations, all citations are weighted equally regardless of their function. This oversimplification is detrimental to the use of citation data in research evaluation methods, according to several researchers [3]. For example, a citation that critiques a paper has a different impact than a citation that serves as a starting point for new research [4], suggesting that the number of citations received is only an

indication of a researcher's productivity and the prominence of his or her work, but is not an indication of the quality of the research itself. This approach to citation analysis, which takes a perfunctory, coarse-grained view, can only reveal superficial citation relationships and does not consider the underlying syntactic and semantic context.

The apprehension about the reliability and accuracy of methods involving the mere counting of citations in the context of research assessment has led to the development of techniques to determine the functional typology of citations. As part of this effort, citation context analysis (CCA) was evolved to uncover the citation phenomenon through a fine-grained and in-depth analysis of citations and to determine the value of each citation based on its context at both syntactic and semantic levels. The premise of CCA is to extract contextual citation information from the citation literature that reflects the content of the citation. When authors of the scientific literature cite a reference, they typically use standard citation descriptions and notations to represent their citations. Citations usually consist of complete sentences, and citation marks are placed at the end of the citation [5]. These types of sentences with obvious citation marks are called explicit citation sentences [6]. However, the citation details of the reference in the cited work are sometimes not limited to the explicit citation sentence but include several surrounding sentences that are often not accompanied by citation marks. Sentences that cite references to the relevant content are called implicit citations [1].

Explicit and implicit citations differ significantly in their identification methods; explicit citations can be easily identified from the citation marks alone [7,8]. In contrast, there are no obvious linguistic and syntactic clues for identifying implicit citations; instead, they must be determined based on their deep semantic features. Therefore, the identification of implicit citations depends primarily on finding the range of implicitly citing sentences around the explicitly citing sentences. Most current studies dealing with citation texts consider only explicit citations and ignore the presence of implicit citations, leaving a large amount of citation information unconsidered. Sometimes a citation is created by inserting a fixed-length text window (containing multiple sentences) around the citation marker. However, this can result in significant amounts of redundancies. Researchers also evaluate the citation text as a variable-length window and use machine learning to determine whether multiple sentences around a citation marker are citation sentences. However, this approach relies heavily on the supervised learning of the text, which requires a large corpus of manually annotated text, making it difficult to generalize to other domains.

The main objective of this study is to develop an unsupervised citation method that identifies implicit citations based on the semantic similarity between citations and the candidate sentences of the cited publication. Using text similarity measures, the sentences around the explicitly cited sentences that are closest to the content of the cited reference are identified as implicit citation sentences. In order to calculate text similarity accurately, various document vector representation models are proposed in this study. The advantage of the proposed approach is that it does not require annotating a large training corpus. Instead, only word vectors from a large unannotated training corpus are used for text similarity computation so the method can be generalized to the scientific literature in numerous domains. Based on the implicit citation sentences identified by the proposed method, a comparative analysis of the differences between explicit and implicit citations in terms of their function and sentiment is performed to demonstrate the utility of identifying implicit citation sentences in performing citation content analysis.

The remainder of this paper is organized as follows. Section 2 reviews the current literature on citation content analysis and classification. Section 3 describes the techniques and methods used in this study to integrate explicit and implicit citation texts and analyze their impact on improving citation analysis and classification. Section 4 presents the experimental setup and case study descriptions used to evaluate the applicability of the proposed model. Section 5 reports the empirical results of this study and discusses their implications for research and practice. Finally, Section 6 presents the conclusions of this work and suggests possible avenues for future research.

## 2. Related Work

In recent decades, a great deal of research has been conducted on citation analysis, with the work of Garfield being the most important. One of the main motivations for studying bibliographic references is to better understand how to evaluate and assess research [8]. Research on citation identification focuses mainly on determining whether a section of text (containing one or more sentences) around a citation marker contains macro-level information, using mainly two types of methods: the fixed-window method and the variable-window method. In the so-called fixed-window method, a fixed-length text window around a citation marker is used as the cited text. However, the main limitation of the fixed-window method is finding the optimal length of the window, which is often measured by the number of words or sentences. To determine the appropriate window length, a number of studies have used cited text in citing documents in information retrieval systems and investigated the effects of the window length of the cited text on the information retrieval result [9–11]. Fixed-window citation detection, while simple and convenient, can involve a significant number of extraneous annotations. An alternative is variable window citation detection, which can be divided into two main types: rule-based and statistical methods. Rule-based methods rely on manually generated cues for citation detection [9,10]. However, due to the different conventions and norms for writing academic literature in different disciplines, this method is less common and not widely used.

Automatic citation classifications based on machine learning methods are the main direction of research in citation-based text prediction. Depending on the machine learning techniques used, they can be further divided into supervised and unsupervised learning. Supervised citation classification models are trained by machine learning algorithms on a corpus of labeled citations to determine whether a sentence belongs to the cited text. Researchers have explored various classification methods, such as the conditional random field (CRF), hidden Markov model (HMM), and support vector machine (SVM). Yousif et al. [1] applied SVM to train a co-reference parsing model from the MUC-7 corpus. They then used the model to identify co-reference links in sentences and identify sentences on the same co-reference link with explicitly cited sentences as implicitly cited sentences [11]; however, the identification performance was not satisfactory. A number of previous studies have trained classification models using syntactic features and citation marker positions. For example, Singh and Paul [12] trained CRF and SVM classifiers from the annotated corpus using a combination of features, such as referent, sentence position, citation marker position, and sentence structure, to identify implicitly cited sentences. The SVM classifier was found to be more effective than the CRF classifier [13]. Angrosh et al. [14] used the trained CRF classifier to detect citations in the literature section of the paper and obtained good results with a recall rate of 96.51%; on the other hand, Yousif et al. [1] used the trained SVM classifier to detect citations in the full text, and the identification results were not satisfactory. Angrosh et al. [14] concluded that implicitly cited sentences are highly correlated with explicitly cited sentences. Therefore, they calculated the probability of their mutual generation based on a linguistic generative model and used it to construct an HMM for citation identification; nevertheless, they achieved a high precision (98.7%) but relatively low recall (50.3%).

While supervised machine learning methods represent the state of the art in citation text identification, the drawback is that a large amount of training data needs to be labeled, and citation text labeling is a difficult task, which hinders the widespread use of such methods. Advances in machine learning and NLP research led to the development of automated methods for the citation context assessment and extraction of textual and non-textual features, followed by citation classification [15]. However, progress in this area has been hampered by the lack of an annotated corpora large enough to generalize the task and which is independent of the domain. In addition, the lack of methods for the formal comparison and evaluation of citation classification systems makes it difficult to assess the state of the art [15]. The domain-specific nature of existing datasets also implies that applying such corpora to multiple disciplines is a rather difficult prospect [16]. In addition,

significant differences in the corpus and in the classification schemes and classifiers used for the experiments signify that reproducing previous results with a new corpus is challenging. The datasets developed for citation classification are highly biased, with the majority of instances falling into the background work, superficial, or neutral categories of citation classification [2,4,16].

In recent years, deep learning techniques have been used for citation classification as there have been advances in this area to solve NLP problems. The main motivation for using neural architectures is their ability to automatically identify features, eliminating the tedious process of defining features before the classification. Popular deep learning approaches for citation classification rely on word representations, such as global vectors for word representation (GloVe), embeddings from language models (ELMo), and bidirectional encoder representations from transformers (BERT) to capture the semantics of citation contexts. Biesialska et al. [17] compared the performance of the Bi-attentive classification network (BCN) and ELMo with the feature-based machine learning approach on the ACL-ARC dataset. The authors emphasize the need for larger datasets to improve the classification performance of deep learning methods. A combined model using the convolutional neural networks (CNN) and LSTM to capture the n-grams and long-term dependencies for multi-task citation functions and sentiment analysis was proposed by Yousif et al. [18]. The latent Dirichlet allocation (LDA) model was used to calculate the similarity between candidate sentences and the cited literature abstracts, and the most similar sentences were considered implicitly cited sentences [16,17]. However, on the one hand, the method has not been evaluated, and it is not known how effective it is; on the other hand, the rigid mapping of an implicitly cited sentence for each citation clearly does not correspond to reality.

There are relatively few relevant studies that address citation text identification using unsupervised learning methods. Zhang et al. [19] used Markov random fields (MRF) for citation text detection. Instead of using training data to train the parameters of the MRF model, the study used the similarity and distance between the candidate sentences and explicitly cited sentences, as well as the lexical features of the candidate sentences to set the model parameters used to generate unsupervised citation text identification; however, the identification results were not satisfactory. Ou and Kim [20] proposed an unsupervised identification method based on text similarity using multiple sentences following the explicitly cited sentence as candidate sentences. Commonly used semantic features include similarity-based indicators. Yasunaga et al. [21] and Tahamtan and Bornmann [6] operationalized this by using cosine similarity. They concluded that this is the best informative feature for classifying the importance of citations. Similarly, in Sahu and Bhowmick [22], the Pearson correlation coefficient between the features and the gold label indicates the effectiveness of the similarity-based features calculated between the title/context of the cited paper and the different aspects of the cited paper.

Compared to supervised learning methods, unsupervised learning methods do not require a training corpus and have a greater potential for application in citation text identification. Therefore, they are a direction worth exploring, as studies investigating the qualitative aspects of citation classification are still in their infancy and need further research. Currently, citation identification research essentially only considers the information in the cited literature, such as the correlation between implicit and explicit citation sentences but ignores the correlation between implicit citation sentences and cited references. The implicit citation sentence, similar to the explicit citation sentence, reflects some aspects of the cited references to some extent. In this work, by manually identifying and analyzing the citation texts in seven cited documents, we found that explicit citation sentences mostly summarize the cited content, while implicit citation sentences further elaborate or evaluate the cited content. In this case, the semantic similarity between the implicit and explicit citation sentences is low, and it is necessary to refer to the information in the cited literature to determine whether the sentence references the content of the cited literature [23].

### 3. Methodology

In this paper, we propose an unsupervised method for identifying implicit citations based on text similarity, as shown in Figure 1. First, the explicit citation sentence is identified based on the citation marker, and then several sentences around the explicit citation sentence are selected as citation candidates. Then, the content similarity between each citation candidate and the citing reference and the cited reference is compared. Finally, the sentence that is most similar to the reference is evaluated as the implicit citation sentence. The similarity in content between the candidate sentences and the cited reference is measured by text similarity, where the content of the cited reference can be represented by the full text or the abstract. To better reflect the similarity between the texts, different methods of representing text vectors were investigated and compared.

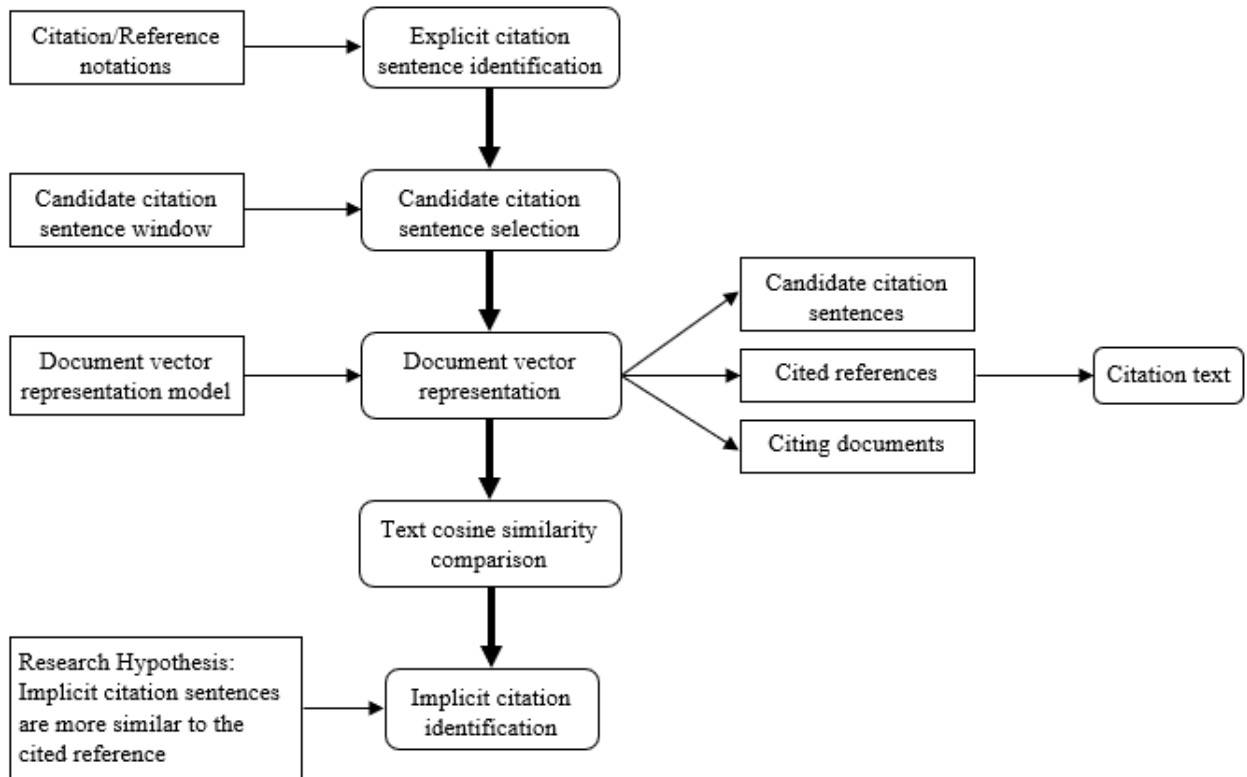

**Figure 1.** Method for identifying implicit citation sentences.

### 3.1. Text Similarity Calculation Based on Different Text Vector Representations

Among the various methods for calculating text similarities, the most commonly used is the vector space model (VSM) based on the term frequency-inverse document frequency (TF-IDF) to represent texts and then calculate the cosine distance between two text vectors. However, this bag-of-words-based text vector representation only considers the co-occurrence of words between texts and does not consider the semantics of the words. This shortcoming is particularly significant in the case of implicit citation detection. The reason for this is that, when citing texts, the authors often refer to the content of the original text of the cited reference by generalization or paraphrasing instead of directly repeating the phrases of the original text, which leads to a significant reduction in the use of the same words, even though the cited content and the original content are still semantically similar.

With the advancement of deep learning techniques, Al-Saqqa and Awajan [24] proposed the CBOW and skip-gram models for word embedding (Word2Vec), which can be used to train word vectors that express the semantics of each word from a large unlabeled corpus. The semantic word vector has made a major breakthrough in computing semantic similarities between words. Based on word vectors, Dinmont et al. [25] proposed two

document embedding (Doc2Vec) models, PV-DM and PV-DBOW. In these two models, a document (or a sentence or a paragraph) is added to all local contexts of a document as a special vocabulary, and then the vector representation of the document is derived using the word embedding model. However, the disadvantage of the Doc2Vec model is that it does not accurately reflect the weights of each vocabulary in the document.

Overall, the vector space model and the deep neural network-based document vector representation model have their own advantages and disadvantages. The vector space model can accurately calculate the weights of words in a document but does not consider the semantic relationships between words. By contrast, the deep neural network model captures the semantics of words but does not consider the weights of words in a document. To accurately compute the semantic similarity between the citation candidates and documents, this paper explores the document vector representation method, which combines the vector space model and the deep neural network model, and proposes two combinations. One is to embed TF-IDF weights into the deep neural network-based document vector representation model. The other is to use the deep neural network-based word vectors in the TF-IDF-based vector space model instead of using the representation of individual words in the original model.

(1) Document vector representation model based on TF-IDF weights and word vectors.

In the traditional vector space model, a document is considered a bag of words consisting of a set of words. For each word in the bag of words, a word vector representation can be trained from a large unlabeled corpus using a deep neural network-based word embedding method. In this paper, we considered a linear weighted combination of word vectors of words to predict the vectors of the bag of words (i.e., documents). However, which combination of weights can more accurately reflect the actual vectors of the bag of words needs further verification. In view of the computationally intensive nature of this validation experiment in a large text corpus and the difficulty of obtaining the actual vector representation of the documents, a multinomial corpus is used to investigate the relationship between different linear combinations of word vectors and bag-of-word vectors.

Following the approach of Nahar et al. [26], this paper also considers a document as a special vocabulary. Since a document can express multiple semantics, this special vocabulary is also a polysemous word. Each semantic meaning of a polysemous word can be considered a special semantic vocabulary, and a polysemous word is a bag of words consisting of all the semantic vocabularies contained within it. To study word vector representation of polysemous words, the SENSEVAL corpus [5] is used in this paper to conduct experiments on the word vector representation of polysemous words. SENSEVAL is a semantic disambiguation corpus of polysemous words created by the Association of Computational Linguistics (ACL) [27]. In this corpus, an example sentence is given for each polysemous word to illustrate the use of the semantics. For example, the polysemous word line has six semantics: cord (rope), division (separation), formation (team), telephone (phone), product (product), and text (text). Consider each of these semantics as a special semantic vocabulary called line_rope, line_division, line_formation, line_phone, line_product, and line_text. Then, the individual words in the example sentence are replaced with the corresponding semantic vocabulary. The training corpus containing the original example sentences and the replaced example sentences is formed, as shown in Table 1. Based on this corpus, Word2Vec [6], a word vector training algorithm developed by Google (Mountain View, CA, USA), is used to train the word vector representation of each polysemous word and semantic vocabulary in the corpus.

**Table 1.** Training corpus of polysemous word vectors (with polysemous word leash as an example).

| Semantic | Example Sentence |
|---|---|
| cord | Original: He managed to land the fish while holding onto his line. |
| | Replacement: He managed to land the fish while holding on to his line rope. |
| division | Original: The legal distinction between commercial and investment banking should be further dissolved. |
| | Replacement: Make the distinction between commercial and investment banking even more hazy. |
| formation | In the passport line at Moscow's Sheremetyevo airport, the correspondent said... |
| | At the Sheremetyevo airport in Moscow, the correspondent said in the development of the passport line... |
| | He placed a second call and returned to the line-phone with the information that . . . |
| phone | He made a second call and returned to the call with the information that . . . |
| | He placed a second call and returned to the line-phone with the information that . . . |
| product | Additionally, Mr. Frashier will advocate for the creation of a line of coating and adhesive products based on proteins. |
| | Additionally, Mr. Frashier will advocate for the creation of a line of protein-based coating and adhesive products. |
| text | Clients apparently receive a one-page bill with one line of text. |
| | Clients apparently receive a one-page bill with a single line of text on it. |

Since a polysemous word can be considered as a bag of words of all the "semantic words" it contains, a linear combination of these "semantic words" and word vectors can be used to compute (predict) the word vectors of polysemous words. In this paper, two linear combination models are defined: the average model ($AWV$ for short), which is the average of the semantic vocabulary word vectors, and the weighted average model ($TF - AWV$ for short), which is the weighted average of the semantic vocabulary word vectors. The mathematical representation of these two models is shown below. Considering the bag of words $D$ is denoted as $D = \{w_1, w_2, \ldots, w_i, \ldots, w_m\}$, where $w_i$ is the $i$th word in the bag of words. The word vector representation of the bag of words $D$, based on the average model, is shown in Equation (1), and the word vector representation of the bag of words $D$, based on the weighted average model, is shown in Equation (2).

$$AWV(D) = \frac{1}{m} \sum_{i=1}^{m} w_i \qquad (1)$$

$$TF - AWV(D) = \frac{1}{\sum_{i=1}^{m} tf_i} \sum_{i=1}^{m} tf_i V_{w_i} \qquad (2)$$

where $V_{w_i}$ is the word vector of the word $w_i$; $tf_i$ is the term frequency ($TF$) weight of the $i$th word $w_i$ in the bag of words $D$.

The predicted word vectors of the polysemantic words are calculated based on the word vectors of the semantic vocabulary using the above two linear combination models. By comparing with the real word vectors of the polysemantic words trained and based on the SENSEVAL corpus, we can determine which linear model can better represent the word vectors of the polysemantic words. Table 2 shows the cosine similarity between the real word vectors of the four polysemantic words and the word vectors of their semantic vocabulary, as well as the predicted word vectors based on the two linear combination models. From Table 2, it can be seen that for the above four polysemantic words (i.e., the bag of words), the word vectors calculated based on the weighted average model are closest to their real word vectors, with a cosine similarity above 0.9 for all of them. This indicates that the frequency-weighted average model can be used to represent the word vectors in the bag of words.

**Table 2.** Cosine similarity between real word vectors of polysemous words and predicted word vectors based on the two linear combination models.

| Polysemy | Semantic Vocabulary | Cosine Similarity | Polysemy | Semantic Vocabulary | Cosine Similarity |
|---|---|---|---|---|---|
| line | line_cord | 0.46 | interest | interest1 | 0.60 |
| | line_devision | 0.55 | | interest2 | 0.63 |
| | line_formation | 0.49 | | interest3 | 0.54 |
| | line_phone | 0.58 | | interest4 | 0.50 |
| | line_product | 0.93 | | interest5 | 0.58 |
| | line_text | 0.47 | | interest6 | 0.87 |
| | AWV | 0.75 | | AWV | 0.79 |
| | TF-AWV | 0.97 | | TF-AWV | 0.93 |
| server | server2 | 0.80 | hard | hard1 | 0.97 |
| | server6 | 0.63 | | hard2 | 0.82 |
| | server10 | 0.80 | | hard3 | 0.62 |
| | server12 | 0.79 | | AWV | 0.95 |
| | AWV | 0.92 | | TF-AWV | 0.98 |
| | TF-AWV | 0.94 | | | |

Although documents are also considered a bag of words, the bag of words in a document differs from the bag of polysemous words mentioned above in one important respect. The words in a document, in addition to the term frequency ($TF$) weight, have a more important weight, the $TF - IDF$ weight, which can more accurately reflect the meaning of the words in the document. Therefore, based on the above term frequency weighted average model, the $TF$ weights of each of these constituent words are replaced with $TF - IDF$ weights to obtain the $TF - IDF$ weighted average model $TFIDF - AWV$ for the document vector prediction, as shown in Equation (3).

$$TF - IDF - AWV(D) = \frac{1}{\sum_{i=1}^{m} tfidf_i} \sum_{i=1}^{m} tfidf_i V_{w_i} \tag{3}$$

where $D$ denotes a document, $wi$ denotes the i-th word in document $D$, $tfidf_i$ denotes the $TF - IDF$ weight of the word $w_i$ in the document $D$, and $V_{w_i}$ is the word vector of word $w_i$.

(2)　Vector space model based on $TF - IDF$ weights and word vectors

When calculating the similarity between the texts based on the traditional spatial vector model, the model considers only the co-occurrence of words in the text. If a word does not occur, its weight is 0. The possible existence of semantically identical or similar alternative words in the text is completely ignored. To address this limitation of the traditional vector space model, this paper uses semantically similar words for mutual substitution and proposes the vector space model $PTFIDF - VSM$ based on $TF - IDF$ weights and word vectors. Given that $V = \{v_1, v_2, \ldots, v_i, \ldots, v_m\}$ is a vector representation of the vector space-based model of document $D$, where $v_i$ denotes the weight of the ith word in the document. If the term occurs in the document, it is assigned its $TF - IDF$ weight; if it does not occur, the $TF - IDF$ weight of the semantically most similar term (often a synonym or near-synonym) that occurs in the document is used instead, but the value is corrected for the semantic similarity between them. This specific calculation is shown in Equations (4) and (5).

$$v_i = \begin{cases} tfidf_i & w_i \in D \\ p_i \times tfidf_i & w_i \notin D \end{cases} \tag{4}$$

$$p_i = \max_{j \neq i} sim\left(V_{w_i}, V_{w_j}\right) \tag{5}$$

where $w_i$ is the *ith* word in the document $D$, and $p_i$ is the semantic similarity between the word most similar to $w_i$ in the document $D$ and $w_i$, which can be calculated using cosine similarity based on the word vector trained by Word2Vec.

In summary, two methods for document vector representation are investigated in this paper, namely the document vector representation model based on $TF-IDF$ weights and word vectors ($TF-IDF-AWV$) and the vector space model based on $TF-IDF$ weights and word vectors ($PTF-IDF-VSM$). In the following, these two models are used for the vector representation of citation candidates and documents for text similarity calculation.

### 3.2. Automatic Identification Method for Implicit Citation Sentences Measurement

According to the document vector representation model proposed in Section 3.1, the cited sentences, cited literature and cited references can be represented as document vectors based on the trained word vectors. The cosine similarity is then used to compare the semantic similarity between the citation sentences and the cited literature and cited references.

(1) Data preparation

To train the word vectors, more than 23,500 articles were first collected randomly from the ACL Anthology Web Corpus (ACL) [28] (https://aclanthology.org/) (accessed on 1 October 2022). The full articles were downloaded and converted to computer-processable text in PDF format using the Apache PDFBox tool developed by Google (Mountain View, CA, USA). Then, the word vectors of all the words were trained using the Word2Vec tool developed by Google (Mountain View, CA, USA).

To compare different document vector models for their implicit citation sentence identification, a sample of papers is collected and manually annotated with all the citation texts contained in them (mainly implicit citation sentences) to build the experimental corpus. While explicit citations are easy to identify based on citation labels, implicit citations are not easy to identify and require reading and understanding the cited references and the citation context of the cited documents. Therefore, identifying all the implicit citation sentences of the cited references (if present) from a paper is a very time-consuming task. In view of this, in this paper, only seven papers published from 2014 to 2017 were randomly selected from three journals [29] in the field of computing, and each citation text in each paper was manually identified to generate a small corpus. The seven papers contained a total of 207 citation texts, among which 139 citation texts (67.1%) contained only explicit citation sentences, while the other 68 citation texts (32.9%) were composed of both explicit and implicit citation sentences, involving a total of 98 implicit citation sentences. Table 3 lists the seven papers selected for building the experimental corpus.

**Table 3.** Sample papers selected for building the experimental corpus.

| Paper | Journal | Explicit Citations | Implicit Citations |
|---|---|---|---|
| Chen, C. [30] | Scientometrics | 26 | 10 |
| Li, K. et al. [31] | Scientometrics | 22 | 14 |
| Pan, W. et al. [32] | Scientometrics | 16 | 21 |
| Chandra, Y., and Walker, R.M. [33] | International Public Management Journal | 21 | 18 |
| Li, P. et al. [34] | Scientometrics | 13 | 11 |
| Olmeda-Gómez, C. et al. [35] | Scientometrics | 24 | 16 |
| Wang, M. et al. [36] | IEEE Access | 17 | 8 |
| Total | | 139 | 98 |

Given the small size of the experimental corpus, which is insufficient to evaluate the final detection effectiveness of the implicit citation sentences, this paper selects two highly cited papers by Jacobs and Hoste [37] and Färber and Jatowt [38], which randomly crawl about 200 citations each, and manually annotates the citation text (including explicit and implicit citation sentences) of each cited paper to build the final evaluation corpus. Since citation styles may differ in different literature areas, the cited papers were crawled from different sources, namely the three databases Scopus, ProQuest, and EBSCOhost.

(2)   Examination of research hypotheses

Compared to the cited literature, the implicitly cited sentences are more similar in meaning to the cited references. Since it is often difficult to obtain the full text of documents in practice, both abstracts and full texts were used to represent the documents in the experiment. Different document vector representation models are used to represent the implicit citation sentence, the cited document (full text or abstract), and the cited reference (full text or abstract), and then the cosine similarity between each implicit citation sentence and the cited document and the mentioned cited reference is compared. The results of the comparison with different document vector representation models are shown in Table 4. It can be seen that more than half of the implicitly cited sentences (at least 57.11%) are more similar to their cited references, regardless of which document vector representation model is used and whether the documents are presented as abstract or full texts.

**Table 4.** Implicit citation sentences based on various document vector representation models with the cited/ reference similarity comparison results.

| Document Vector Representation Model | Acronym | Proportion of Implicitly Cited Sentences That Are More Similar to the Cited Reference | |
| --- | --- | --- | --- |
| | | Literature (Documents) Represented as Abstract | Literature (Documents) Represented as Full Text |
| Traditional vector space model | TF-IDF-VSM | 69.4% | 57.12% |
| Document vector representation model based on TF or TF-IDF weights and word vectors | TF-AWV | 97.38% | 59.21% |
| | TF-IDF-AWV | 80.33% | 70.83% |
| Vector space model based on TF-IDF weights and word vectors | PRFIDF-VSM | 73.63% | 70.66% |

Using the document vector representation model based on TF-IDF weights and word vectors, and literature abstracts, the effect is more significant, with more than 80.33% of the implicitly cited sentences appearing more similar to the cited references. Moreover, the degree of similarity between the implicitly cited sentences and the cited references is more pronounced when the document abstracts (both cited and cited references) are used instead of the full text. This is because the cited sentences tend to summarize the content of the cited references, and the summary is also general, so the semantic similarity between the cited sentences and the summary is higher than that of the full text. The experimental results show that the implicit method proposed in this paper for identifying citation sentences based on text similarity is extremely useful and feasible.

(3)   Range determination of candidate citations

The main problems with the text similarity-based method for identifying implied citation sentences proposed here are twofold. The first is determining the range of citation sentences to consider. If the range chosen is too large, it will result in high noise and, thus, low accuracy. If the range chosen is too small, the truly implicitly cited sentences will be overlooked, resulting in low accuracy. Second, the vector representation of the documents directly affects the accuracy of the text similarity computation. In this paper, two models for the document vector representation are proposed, and it needs to be determined which model has a better detection performance.

In this paper, the range of citation-eligible sentences is determined by the experimental corpus. The range of citation-eligible sentences has two windows on the left and right sides, where the left window refers to the sentences before the explicitly cited sentences, and the right window refers to the sentences after the explicitly cited sentences. The lengths of the two windows are set relatively independently. The effect of changing the length of the left and right windows on the detection result is examined separately by using the F1 value as the evaluation index. First, the length of the right window is set to 10 (i.e., 10 sentences after the explicit citation), and the length of the left window is set from 1 to

9 (i.e., 1–9 sentences before the explicit citation), and the sentences in the left and right windows are used as citation candidates. The detection result for the different lengths of the left window is shown in Figure 2. The different curves show the results of using different models of document vector representation. It can be seen that for all the document vector representation models, the F1 value for detecting the sentences with implicit citations is highest when the length of the left window is 2. Therefore, the length of the left window is set to 2 for the range of candidate citations. Next, the length of the left window is set to 2, and the length of the right window is adjusted from 1 to 14.

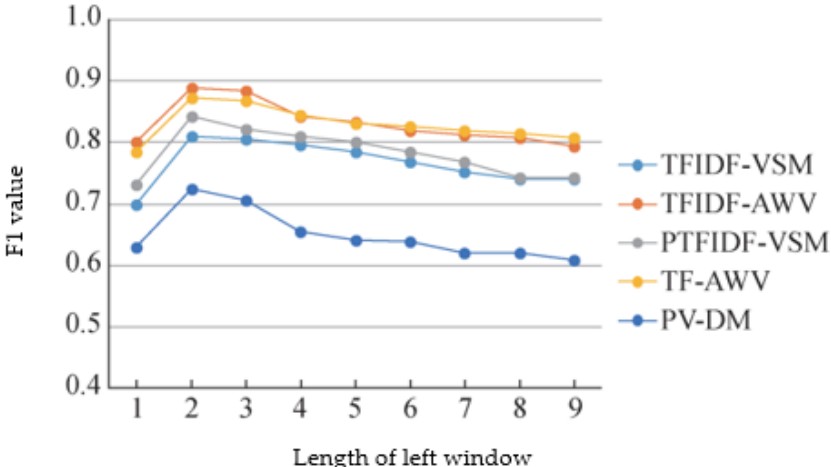

**Figure 2.** Variation in implicit reference sentence detection performance with the length of the left window (right window length is fixed at 10).

The detection results with different lengths of the right window are shown in Figure 3. It can be seen that the F1 value gradually increases as the length of the right window increases for all the document vector representation models. When the length of the right window is 10, the increase in the F1 value flattens out and no longer changes significantly. Therefore, the length of the right window for the range of candidate sentences is set to 10. The range of candidate citations was set to the two sentences before the explicit citation sentence and the 10 sentences after.

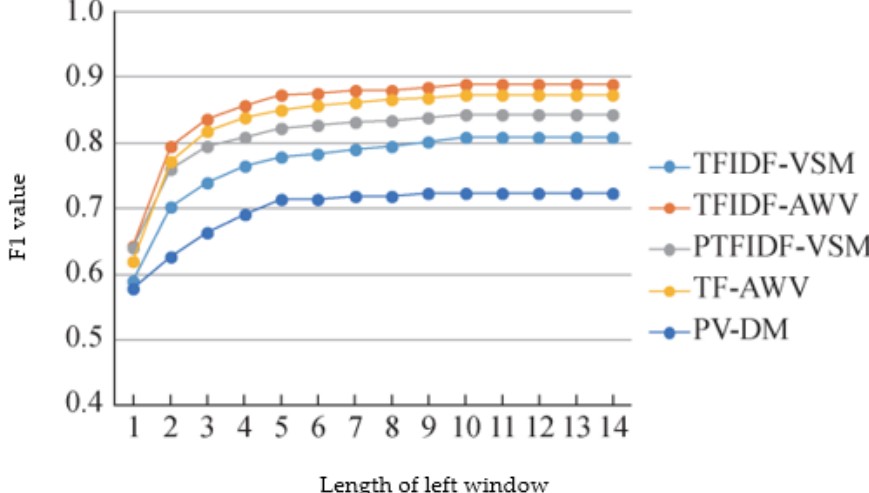

**Figure 3.** Implicit reference sentence detection performance as a function of right window length (left window length fixed at 2).

(4) Evaluation and Analysis of Implicit Citation Sentence Detection Based on Different Document Vector Representation Models

After determining the range of candidate citation sentences, the experimental corpus is used to evaluate the performance of the two document vector representation models proposed in this paper for implicit citation sentence detection. As a comparison, the traditional vector space model and the Doc2Vec document vector representation model [39] are used as benchmark models.

Table 5 shows the performance of implicit citation sentence detection based on various document vector representation models, evaluated by precision (P), recall (R), and F1 values. It can be seen that the model with the best detection performance is the document vector model based on TF-IDF weights and word vectors, with F1 values of 88.87% and 82.43% for the abstract and full text of the documents, respectively. When the abstract is used to represent the literature, the identification is better than when the full text is used. In reality, the abstracts of documents are far easier to obtain than full texts, so this finding is very significant for practical applications. For all the document vector representation models, the detection accuracy was very high; all of them reached more than 80%, and some of them even reached more than 99%, but the detection recall rate is not satisfactory, with the highest reaching only 80%. This indicates that some implicit references are still missed; therefore, the key to improving the overall performance of detection is to increase the recall rate.

**Table 5.** Performance of implicit citation sentence detection based on various document representation models.

| Document Vector Representation Model | Acronym | Literature (Documents Are) Is Represented as an Abstract | | | Literature (Documents Are) Is Represented as Full Text | | |
|---|---|---|---|---|---|---|---|
| | | P% | R% | F1% | P% | R% | F1% |
| Traditional vector space model | TF-IDF-VSM | 97.26 | 69.34 | 80.96 | 99.65 | 57.12 | 72.71 |
| Doc2Vec model | PV-DBOW | 84.97 | 63.07 | 72.40 | 97.67 | 54.29 | 69.78 |
| Document vector model based on TF | TF-AWV | 96.53 | 97.79 | 87.37 | 99.26 | 59.21 | 74.17 |
| or TF-IDF weights and word vectors | TFIDF-AWV | 99.44 | 80.33 | 88.87 | 98.58 | 70.83 | 82.43 |
| Vector space model based on TF-IDF weights and word vectors | PTFIDF-VSM | 98.77 | 73.63 | 84.37 | 100.00 | 70.66 | 82.81 |

In order to improve the recall rate, detection methods based on different document vector representation models are combined. First, the first document vector representation model is used to identify the citation and non-citation sentences from candidate citation sentences; next, the non-citation sentences filtered out in the first step are identified by the second document vector representation model, from which the missing implicit citation sentences are identified. Table 6 shows the performance of different combination modes for identifying implicitly cited sentences. It can be seen that the implicitly cited sentence detection based on the combined model can further improve the recall rate and greatly improve the overall performance of detection. The best combination model is the combination of the document vector model (TFIDF-AWV) based on TF-IDF weights and the word vectors and the vector space model (PTFIDF-VSM) based on TF-IDF weights and word vectors. When the abstract representation of documents is used, the F1 value reaches over 94%. The order of the combination of these models has a slight effect on the detection performance, but the difference between them is not significant and negligible.

**Table 6.** Performance of implicit citation sentence detection based on different combination models.

| Combination Model | Literature Represented as an Abstract | | | Literature Represented as Full Text | | |
|---|---|---|---|---|---|---|
| | P% | R% | F1% | P% | R% | F1% |
| TF-IDF-AWV + PTFIDF-VSM | 99.46 | 90.52 | 94.78 | 99.06 | 87.64 | 92.98 |
| PT-FIDF-VSM + TF-IDF-AWV | 98.91 | 90.11 | 94.31 | 99.53 | 88.05 | 93.44 |
| TF-IDF-AWV + PV-DBOW | 96.68 | 89.55 | 92.98 | 97.93 | 80.74 | 88.50 |
| TF-IDF-AWV + TF-IDF-VSM | 98.48 | 87.79 | 92.83 | 98.77 | 80.50 | 88.70 |

(5)  Final evaluation of implicit citation sentence detection based on the best combination model

In order to evaluate the final detection effect of implicitly quoted sentences, the best detection model (i.e., the combined model TF-IDF-AWV + PTFIDF-VSM) is used in this paper to automatically recognize the implicitly quoted sentences of two highly cited papers on the evaluation corpus. Both the compared cited and cited literature were used for the abstracts, and the results are shown in Tables 7 and 8, respectively. The experimental results show that the overall effect of the implicit citation sentence detection of both highly cited papers is satisfactory, with F1 values as high as 92.2%, indicating that the implicit citation sentence detection method proposed in this paper is very effective. By comparison, the precision rate of implicit citation sentences in the highly cited papers of the deep neural network (89.2%) is lower than that of the highly cited papers of the LDA topic model (96.8%), but the recall rate (96.6%) is higher than that of the latter (88.5%). There is no significant difference in the effectiveness of identifying implicitly cited sentences in the cited papers of different fields [40].

**Table 7.** Implicit citation sentence detection results of highly cited papers with deep neural network.

| Citation Field | Number of Cited Articles | Number of Explicit Citations | Number of Implicit Citations | Implicit Citation Identification Results | | |
|---|---|---|---|---|---|---|
| | | | | P% | R% | F1% |
| Computer science | 90 | 119 | 215 | 89.7 | 97.8 | 93.6 |
| Engineering | 66 | 87 | 137 | 91.2 | 96.1 | 93.6 |
| Management | 26 | 41 | 54 | 89.4 | 98.0 | 93.5 |
| Physics | 25 | 32 | 55 | 90.0 | 93.2 | 91.5 |
| Medicine | 23 | 33 | 49 | 81.1 | 95.2 | 87.6 |
| Other | 22 | 34 | 69 | 96.5 | 88.3 | 92.2 |
| Total | 253 | 313 | 579 | - | - | - |
| Average | - | - | - | 89.56 | 94.66 | 91.91 |

**Table 8.** Implicit citation sentence detection results of highly cited papers for LDA topic model.

| Citation Field | Number of Cited Articles | Number of Explicit Citations | Number of Implicit Citations | Implicit Citation Identification Results | | |
|---|---|---|---|---|---|---|
| | | | | P% | R% | F1% |
| Computer science | 93 | 145 | 254 | 97.4 | 88.8 | 92.9 |
| Engineering | 40 | 59 | 90 | 96.4 | 87.9 | 91.9 |
| Management | 29 | 42 | 83 | 95.6 | 87.1 | 91.2 |
| Medicine | 14 | 23 | 46 | 95.4 | 88.5 | 91.8 |
| Business | 11 | 12 | 23 | 100.0 | 91.0 | 95.3 |
| Other | 26 | 48 | 91 | 96.4 | 88.2 | 92.1 |
| Total | 213 | 333 | 587 | - | - | - |
| Average | - | - | - | 96.9 | 88.6 | 92.5 |

## 4. Comparative Analysis of Explicit and Implicit Citation Sentences

The citation classification problem from a discourse analyst point of view was later studied by Brookes and McEnery [40], Hjelm [41], and Jacobs, G., and Hoste [42]. Here, the explicitly mentioned words or phrases surrounding the citation are analyzed to interpret the author's intentions for citing a document [43]. To this end, several taxonomies, from the very generic to the more fine-grained, were developed, reflecting on citation types from a range of perspectives [44]. These include understanding citation functions, which constitute the roles or purposes associated with a citation, by examining the citation context [4]; citation polarity or sentiment, which gives insight into the author's disposition towards the cited document [45]; and citation importance, where the citations are grouped based on how influential/important they are to the cited document [6,26].

The current research on citation content analysis focuses on the citation function (or motivation), citation sentiment, and citation theme of the citation text. The citation text consists of a combination of explicit citation sentences and their surrounding implicit citation sentences. However, it is not yet known whether there are differences between these two types of citation sentences in expressing the author's citation content. In this paper, we still use Yousif et al.'s [1] paper as an example to analyze the difference between explicit and implicit citation sentences in expressing citation function and sentiment when this highly cited paper is cited by other literature.

To construct the analytic corpus, the full text and abstracts of 1203 scientific papers citing the literature were first obtained from the Elsevier website. Then, the citation text identification method proposed in this paper was used (i.e., explicit citation sentences were identified using citation markers, and then the combination of the TF-IDF-AWV + PTFIDF-VSM model was used to identify the implicit citation sentences around them based on text similarities in the literature abstracts). A total of 1633 cited texts were identified from these cited documents, including 1633 explicit citations and 3574 implicit citations, for a total of 5196 cited sentences.

### 4.1. Comparative Analysis of Citation Functions

According to Will [46] classification criteria and citation, functions were divided into four categories stating the background of the study ("background" category), providing the technical basis ("use" category), stimulating the basic ideas of existing studies ("based on" category), and comparing existing research with other research ("comparison" category). Using the automatic citation function classification tool developed by Daradkeh et al. [44], the citation functions of all the citation sentences (both explicit and implicit) were automatically classified. The distribution of different citation functions in the two types of citation sentences is shown in Table 9.

**Table 9.** Distribution of different citation functions in explicit citation sentences and implicit citation sentences.

| Citation Category | (Background) Category | | (Use) Category | | (Based on) Category | | (Comparison) Category | |
|---|---|---|---|---|---|---|---|---|
| | No. | % | No. | % | No. | % | No. | % |
| Explicit citations | 1224 | 75.5 | 307 | 19.9 | 61 | 3.7 | 34 | 2.4 |
| Implicit citations | 2763 | 77.6 | 756 | 22.2 | 13 | 0.4 | 35 | 1.0 |

The chi-square test was used to test whether the distribution of explicit and implicit citation sentences in terms of the citation function was consistent, i.e., to test whether there was a correlation between the citation sentence category and the citation function category. According to the statistical results, the chi-square value is 104.5, which is greater than the critical value (8.06 for a degree of freedom of three and a confidence level of 99.96), indicating that there is a significant difference in the distribution of the citation function between the explicit and implicit citation sentences. The proportion of citation functions of "context" and "use" was slightly higher in the implicit citation sentences than in the explicit

citation sentences. On the contrary, the proportion of "based on" and "comparison" was significantly higher in the explicit than in the implicit quotations.

*4.2. Comparative Analysis of Citation Sentiment*

According to the currently prevailing citation sentiment classification criteria, citation sentiment was classified into three categories: positive citation, negative citation, and neutral citation. The automatic citation sentiment classification tool developed by Daradkeh et al. [47] was also used to automatically classify the citation sentiment of all the citation sentences (both explicit and implicit). The distribution of different citation sentiments in the two types of citation sentences is shown in Table 10.

**Table 10.** Distribution of different citation sentiments in explicit and implicit citation sentences.

| Citation Category | Positive Citation | | Negative Citation | | Neutral Citation | |
|---|---|---|---|---|---|---|
| | No. | % | No. | % | No. | % |
| Explicit citation sentence | 735 | 45.4 | 84 | 5.2 | 806 | 49.7 |
| Implicit citation sentence | 547 | 15.4 | 209 | 5.9 | 2810 | 78.9 |

Using the same chi-square test for the data in Table 9, a chi-square value of 542.0 was obtained, which is greater than the critical value (5.99 at a degree of freedom of two and a confidence level of 99.95%). This indicates that there is a significant difference in the distribution of the citation sentiment between the explicit and implicit citation sentences. The percentage of positive citations in the explicit citation sentences (45.4%) was very prominent and much higher than that of the implicit citation sentences (15.4%); close to 81% of the implicit citation sentences were neutral citations. This indicates that explicit citation sentences provide more subjective evaluations of the cited references, while implicit citation sentences provide more objective descriptions of the references.

## 5. Limitations and Future Research

First, this paper only identifies citation texts at the macro level. Future research will focus on micro-level citation text detection, that is, the further precise identification of text fragments (e.g., phrases or clauses) related to the content of a given reference from explicitly cited sentences with citation labels.

Second, the researchers recruited to annotate the citation context sample are early professionals, as their expertise on the subject is still nascent. It would be desirable to involve more experienced individuals in the annotation of the data. In addition, the inter-rater agreement was below 70%, and the Cohen Kappa score for 139 was poor (0.27). This is a limiting factor, as the lower inter-rater reliability scores reflect the poor reliability of the data collection instrument. However, the impact of poorer coder reliability can be mitigated by assessing only data points on which both annotators agree rather than assessing other records on which they disagree. In addition, coding such a broad and multidisciplinary text may be more challenging than coding texts from narrower or more specific disciplines/subdisciplines, such as machine learning, text mining, and sentiment analysis.

Third, only sentence-level citation texts were identified, and phrase-level citation text detection needs to be further explored. In this analysis, only articles from the ACL Anthology Web Corpus (ACL) [28] were evaluated. This may have an impact on the external validity of the paper's conclusions. The results may change if the corpora from other disciplines are evaluated, as these disciplines have different methods of citation. Likewise, the half-life of publications may affect the remaining citation patterns of the articles. In the case of shorter half-lives, the residual citation characteristics may differ significantly over successive generations.

## 6. Conclusions

This paper focuses on the identification of citation texts, with an emphasis on identifying implicit citation sentences without citation labels. An unsupervised implicit citation sentence detection method based on text similarities is proposed. By comparing the text similarity, the sentences around the explicit citation sentences that are more similar to the content of the cited references are identified as implicit citation sentences. In order to calculate the text similarity accurately, different document vector representation models are proposed in this paper. Additionally, through experimental comparison, the best combination model of the document vector model based on TF-IDF weights and word vectors and the vector space model based on TF-IDF weights and word vectors is derived. Using this model to automatically identify implicitly cited sentences in some of the cited documents of two highly cited papers, the F1 value reached over 93%, which illustrates the effectiveness of the method in this paper. Compared with existing citation text identification methods, the advantage of this method is that no annotated corpus is required, and only word vectors are trained from a large-scale unannotated training corpus to provide a basis for the text similarity calculation.

Based on the identification of explicit and implicit citations, a comparative analysis of the citation contents expressed in the explicit and implicit citations was conducted. In terms of the citation function, the results showed that implicit citations were mostly phrased for explaining the background of the study and providing a technical basis, while explicit citations were mainly phrased for stimulating research ideas and comparing with other studies. In terms of citation emotion, implicit citation sentences tended to objectively describe the citation content of the references and were mostly neutral, while explicit citation sentences contained more positive evaluations of the cited literature and had an obvious sentimental tendency. In view of the differences between the two in terms of their expressive content, it is necessary to include implicit citation sentences in the identification and analysis of citation texts because an implicit citation sentence is more similar in content to the cited reference than to the citation literature in which it is located [48].

**Author Contributions:** Conceptualization, M.D.; methodology, M.D. and R.M.; software, R.M.; validation, M.D. and R.M.; formal analysis, A.E.-H.; investigation, A.E.-H.; writing—original draft preparation, M.D.; writing—review and editing, P.P.; supervision, P.P.; project administration, M.D., R.M. and P.P. All authors have read and agreed to the published version of the manuscript.

**Funding:** This research received no external funding.

**Data Availability Statement:** Not applicable.

**Conflicts of Interest:** The authors declare no conflict of interest.

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
