# Peer review of "A Semantic Similarity-Based Identification Method for Implicit Citation Functions and Sentiments Information"

_information, doi:10.3390/info13110546_

Round 1
Reviewer 1 Report
In this paper, the authors propose an unsupervised citation detection method that uses the semantic similarity between citations and candidate sentences to identify implicit citations, determine their function, and analyze their sentiment. They also validate the model for identifying implicit citations, deep neural networks, and LDA topic modeling on two citation datasets.
The problem addressed in this paper is interesting and worth exploring. Therefore, I think it deserves consideration for publication in the Journal of Information.
However, I have a few minor corrections that need to be addressed.
1- The references should be revised because there are some missing parts (e.g., issues, volume numbers, etc.).
2- The authors should add more text in the "Limitations" column to emphasize the limitations of this study.
3- Please add more references from the sources as it is appropriate to link your work with some related papers from the same journal.
Author Response
Point 1: The references should be revised because there are some missing parts (e.g., issues, volume numbers, etc.).
Response:
References have been revised and updated.
Point 2: The authors should add more text in the "Limitations" column to emphasize the limitations of this study.
Response:
The limitations section has been expanded and new text has been added
Point 3: Please add more references from the sources as it is appropriate to link your work with some related papers from the same journal.
Response:
References have been revised and updated.
Reviewer 2 Report
The aim and approach in this paper are of interest. However, there are a number of minor errors in the description, and it does not appear to have been sufficiently polished. At least, the authors will need to correct the following points found by the reviewer.
P5, L2 in Sec. 3) "Figure 1" would be correct instead of "Figure 2".
P6, 2nd para. in Sec. 2.1.1) It would be "line" instead of "leash".
P7, 3rd para. in Sec. 2.1.1) Why is it named "TF-AWV"? It is easy to guess, but please specify what "TF" stands for.
Eq. (1) ) "V_wi" would be correct instead of "wi".
After Eq. (2) ) "The term frequency" would be more appropriate than "the word frequency" since it is widely recognized as a technical term in this field.
Table 2) The table is not properly divided for the third and fourth words ("interest" and "hard"). Please correct it.
P8, 1st para.) "TFIDF-AWV" would be correct instead of "TFIDFAWV".
After Eq. (3) ) A hyphen is missing in "TFIDF".
Eq. (4) ) In the second line it would be "wi \notin D".
Eq. (5) ) It would be "Sim(V_w_i, V_w_j)" (The first "w_i" should be a subscript).
P8, 1st line in Sec. 3) It would be "Section 2.1".
P9, 2nd para. in Sec. 3.1) To increase the credibility and reproducibility of the papers, the bibliography of the selected seven papers should be listed in a table.
P9, at the enc of 2nd para. in Sec. 3.1) The calculation seems strange. Where did the number 98 come from?
The sentence after Eq. (5) ) "where, ..." without paragraph break would be better instead of "Here, ". Moreover, "w_j is the j-th ..." is redundant.
In Table 3) The 97.38% in the table is the highest value. I don't know why the 80.33% is written in bold.
P10, 3rd line from the bottom) I don't understand why this sentence is connected with "Previously, ".
P11, 2nd para. in Sec. 3.4) The order of P, R, and F1 in this statement differs from the order of the columns in the table.
In Table 4) It would be "TFIDF-AWV" instead of "PTIDF-AWV". Moreover, it is strange to bold 82.43 because 82.81% is greater than 82.43%.
P12, lines 5-8 in the 1st para.) Abbreviations should be explained where they first appear. At least, it is already used in Table 4.
P12, 3rd line of 1st para. in Sec. 3.5) I do not know what is "two". The tables appear to list 6 areas.
Almost all tables) It is very difficult to see the table spanning two pages. Please reconsider their layout.
Overall) Formulas in paragraphs should also be written in formula mode (in italics).
Author Response
Point 1: P5, L2 in Sec. 3) "Figure 1" would be correct instead of "Figure 2".
Response: Thank you for your comments. In the revised paper.
Point 2: P6, 2nd para. in Sec. 2.1.1) It would be "line" instead of "leash".
Response: Thank you for your suggestion. I have thoroughly checked and proofread the paper to make sure there are no typos or grammatical errors in the paper.
Point 3: P7, 3rd para. in Sec. 2.1.1) Why is it named "TF-AWV"? It is easy to guess, but please specify what "TF" stands for.
Response: We updated text to specify what "TF" and "TF-AWV" stand for.
Point 4: Eq. (1) ) "V_wi" would be correct instead of "wi".
Response: Vwi has been corrected as suggested by reviewer 2.
Point 5: After Eq. (2) "The term frequency" would be more appropriate than "the word frequency" since it is widely recognized as a technical term in this field.
Response: The word frequency was changed to term frequency in all parts of the paper as suggested by reviewer 2.
Point 6: Table 2) The table is not properly divided for the third and fourth words ("interest" and "hard"). Please correct it.
Response: The format of the table has been corrected. The table is now divided correctly.
Point 7: P8, 1st para.) "TFIDF-AWV" would be correct instead of "TFIDFAWV".
Response: TFIDF-AWV has been corrected as suggested by reviewer 2.
Point 8: After Eq. (3) ) A hyphen is missing in "TFIDF".
Response: hyphen has been added for "TFIDF" as recommended by reviewer 2
Point 9: Eq. (4) ) In the second line it would be "wi \notin D".
Response: corrected
Point 9: Eq. (5) ) It would be "Sim(V_w_i, V_w_j)" (The first "w_i" should be a subscript).
Response: corrected
Point 10: P8, 1st line in Sec. 3) It would be "Section 2.1".
Response: corrected
Point 11: P9, 2nd para. in Sec. 3.1) To increase the credibility and reproducibility of the papers, the bibliography of the selected seven papers should be listed in a table.
Response: Table inserted
Point 12: P9, at the enc of 2nd para. in Sec. 3.1) The calculation seems strange. Where did the number 98 come from?
Response: Calculated manually by the paper authors.
Point 13: The sentence after Eq. (5) ) "where, ..." without paragraph break would be better instead of "Here, ". Moreover, "w_j is the j-th ..." is redundant
Point 14: In Table 3) The 97.38% in the table is the highest value. I don't know why the 80.33% is written in bold.
Response: Corrected
Point 15: P10, 3rd line from the bottom) I don't understand why this sentence is connected with "Previously, ".
Response: The word "Previously" has been deleted
Point 16: P11, 2nd para. in Sec. 3.4) The order of P, R, and F1 in this statement differs from the order of the columns in the table
Response: Order of P, R, and F1 has been corrected
Point 17: In Table 4) It would be "TFIDF-AWV" instead of "PTIDF-AWV". Moreover, it is strange to bold 82.43 because 82.81% is greater than 82.43%.
Response: corrected
Point 18: P12, lines 5-8 in the 1st para. Abbreviations should be explained where they first appear. At least, it is already used in Table 4.
Point 19: P12, 3rd line of 1st para. in Sec. 3.5) I do not know what is "two". The tables appear to list 6 areas.
Response: We deleted the word "two"
Point 20: Almost all tables) It is very difficult to see the table spanning two pages. Please reconsider their layout.
Point 21: Overall) Formulas in paragraphs should also be written in formula mode (in italics).
Thank you for highlighting this important issue. In the revised paper.